# Preparation of High Refractive Index Composite Films Based on Titanium Oxide Nanoparticles Hybridized Hydrophilic Polymers

**DOI:** 10.3390/nano9040514

**Published:** 2019-04-02

**Authors:** Makoto Takafuji, Maino Kajiwara, Nanami Hano, Yutaka Kuwahara, Hirotaka Ihara

**Affiliations:** 1Department of Applied Chemistry and Biochemistry, Kumamoto University, 2-39-1 Kurokami, Chuo-ku, Kumamoto 860-8555, Japan; 181d8811@st.kumamoto-u.ac.jp (M.K.); Nanami_Hano@kumadai.jp (N.H.); kuwahara@kumamoto-u.ac.jp (Y.K.); 2Kumamoto Institute for Photo-Electro Organics (PHOENICS), 3-11-38 Higashimachi, Higashi-ku, Kumamoto 862-0901, Japan

**Keywords:** titanium oxide, hybrid material, anatase, organic-inorganic hybrid, nanocomposite, nanoparticles, high refractive index material

## Abstract

Optical materials with high refractive index (n) have been rapidly improved because of urgent demands imposed by the development of advanced photonic and electronic devices such as solar cells, light emitting diodes (LED and Organic LED), optical lenses and filters, anti-reflection films, and optical adhesives. One successful method to obtain high refractive index materials is the blending of metal oxide nanoparticles such as TiO_2_ and ZrO_2_ with high n values of 2.1–2.7 into conventional polymers. However, these nanoparticles have a tendency to agglomerate by themselves in a conventional polymer matrix, due to the strong attractive forces between them. Therefore, there is a limitation in the blending amount of inorganic nanoparticles. In this paper, various hydrophilic polymers such as poly(N-hydroxyl acrylamide) (*p*HEAAm), poly(vinyl alcohol), poly(ethylene glycol), and poly(acrylic acid) were examined for preparation of high refractive index film based on titanium oxide nanoparticle (TiNP) dispersed polymer composite. The hydrogen bonding sites in these hydrophilic polymers would improve the dispersibility of inorganic nanoparticles in the polymer matrix. As a result, *p*HEAAm exhibited higher compatibility with titanium oxide nanoparticles (TiNPs) than other water-soluble polymers. Transparent hybrid films were prepared by mixing *p*HEAAm with TiNPs and drop casting the mixture onto a glass plate. The refractive indices of the films were in good agreement with calculated values. The compatibility of TiNPs with *p*HEAAm was dependent on the surface characteristics of TiNPs. TiNPs with the highest observed compatibility could be hybridized with *p*HEAAm at concentrations of up to 90 wt%, and the refractive index of the corresponding film reached 1.90. The high compatibility of TiNPs with *p*HEAAm may be related to the hydrophilicity and amide and hydroxyl moieties of *p*HEAAm, which cause hydrogen bond formation on the TiO_2_ surface. The obtained thin film was slightly yellow due to the color of the original TiNP dispersion; however, the transmittance of the film was higher than 80% in the wavelength range from 480 to 900 nm.

## 1. Introduction

High refractive index materials are required to improve the performance of eyeglass lenses, optical fibers, and optical devices, such as waveguide-based optical circuits, optical interference filters and mirrors, optical sensors, and solar cells [1,2]. High refractive index polymers and/or polymer composites have been attracting much attention as they are light-weight and have high flexibility and high formability compared to inorganic materials. The most useful applications for polymer-based high refractive index materials are as optical data storage [3], lenses [4], anti-reflective coatings [5] and immersion lithography [6]. The recent developments have been summarized in review articles [7,8] and the references therein. The refractive index (n) values of conventional polymers are lower (n = 1.3–1.7) [9] than those of inorganic materials [10,11]. Several methods to obtain high refractive index materials have been previously published. These methods can be categorized into two approaches; one is the incorporation of heavy atoms such as sulfur and/or a halogen into the polymer [12], and the other is the fabrication of polymer composites with high refractive index inorganic or metal nanoparticles (NPs) [13,14,15,16,17,18,19,20,21,22]. There are some challenges to increasing the refractive index in each approach. In the first case, it is technically difficult and costly to introduce heavy atoms into the polymer backbone. The maximum refractive index reported for a material prepared using this approach is 1.84, which was obtained for poly(sulfur-random-(1,3,5-triisopropenylbenzene)) [23]. In the second case, a variety of NPs such as those of zirconium oxide (ZrO_2_) [13], titanium oxide (TiO_2_) [14,15,16,17,18], alumina oxide [19], gold [20], and nanodiamond [21,22] have been used as nano-sized fillers. However, inorganic NPs agglomerate in the polymer matrix due to their high surface energies and low compatibilities with the polymer. Therefore, comparably higher amount of inorganic nanoparticles-involving polymer composites could be prepared by in situ polymerization of polymerizable monomers in the presence of inorganic nanoparticles [24,25], or in-site sol-gel reaction of inorganic precursors in the presence of a polymer matrix [26,27]. The polymer composites with higher amounts of titania NPs showed higher refractive indices of more than 1.7. For instance, poly(4-vinylbenzyl alcohol) was reported as a matrix polymer for titania NPs. The composites were prepared by in-site polymerization of 4-vinylbenzyl alcohol in the acid surface-modified TiO_2_ nanoparticles dispersion. TiO_2_ nanoparticles were dispersed in the polymer at up to 60 wt%, and the refractive index of the composite reached 1.77 [18]. The hydroxyl groups in the side chains of polymers may play an important role in the dispersion of the titania NPs into the polymer matrix. Recently we have reported a new method for fabrication of high refractive index materials based on simple blending of polymers and heteropoly acids [28,29]. In these cases, hydrophilic moieties of polymers such as carbonyl and hydroxyl groups exhibited high compatibility with heteropoly acids which were molecularly dispersed in the polymer matrix. The obtained hybrid showed a dramatic increase in the refractive index of the general polymer, while maintaining a high transparency. Based on these results, we focus on hydrophilic polymers as a matrix for polymer composites containing titania NPs. In this study, the hydrophilic polymers are examined as a matrix for titania NPs, and the optical properties including refractive index of the obtained composites are evaluated.

## 2. Materials and Methods

### 2.1. Materials

Poly(*N*-hydroxyethyl acrylamide) (*p*HEAAm) was synthesized by thermally initiated radical polymerization (see Appendix A). Poly(acrylic acid) (*p*AA, Mw = 5000) and poly(vinyl pyrrolidone) (*p*VP, Mw = 36,000) were purchased from FUJIFILM Wako Pure Chemical Corporation (Osaka, Japan) and Honeywell Fluka (Morristown, NJ, USA), respectively, and poly(ethylene glycol) (*p*EG, Mw = 20,000) and poly(vinyl alcohol) (*p*VA, Mw = 500) were purchased from Nacalai Tesque (Kyoto, Japan). In this study, three different aqueous dispersions of titanium oxide nanoparticles (TiNP-x) were used for fabricating polymer composites. TiNP-1 (40 nm, 0.85 wt%, PTA) and TiNP-2 (20 nm, 0.85 wt%, TPX-HP) were purchased from Kon Corporation (Saga, Japan), and TiNP-3 (70 nm, 18.1 wt%, TisolA) was kindly provided by NYACOL Nano Technologies Inc. (Ashland, MA, USA). Deionized water was prepared with an Elix UV 3 (Merck KGaA, Darmstadt, Germany) and used as a solvent.

### 2.2. Preparation of Composites

The aqueous mixtures of polymers and TiNPs were prepared by mixing an aqueous solution of polymer and TiNP-x in predetermined ratios. Thin films of polymer/TiNP composites were prepared by drop casting 300 µL of the aqueous mixture onto glass and subsequently allowing it to dry either slowly, at 10 °C, for 24 h, or quickly, at 50 °C, for 1 h. The glass was treated with aqueous 1 mol L^−1^ sodium hydroxide solution before casting. The compatibilities of TiNPs with hydrophilic polymers were observed at 70 wt% TiNP.

### 2.3. Measurements

X-ray diffraction measurements (XRD) of TiNPs were performed on a SmartLab (Rigaku corporation, Tokyo, Japan) operating in reflection mode with CuKα radiation (λ = 1.54 Å, 45 kV, 200 mA) and a diffracted beam monochromator, using a step scan mode with a step of 0.02° (2θ) and 4 s per step. The morphologies of TiNPs were observed using transmittance electron microscopy (TEM) using a JEM-1400Plus (JEOL, Tokyo, Japan). Dynamic light scattering (DLS) measurements were performed with a Zetasizer Nano ZS (Malvern, Tokyo, Japan). Thermogravimetric (TG) analyses were carried out with TG/DTA-6200 (Hitachi-Hitech, Tokyo, Japan) under air at a flow rate of 200 mL/min and a heating rate of 10 °C/min.

The refractive index values and thicknesses of the films were evaluated using a prism coupler SPA-4000 (Sairon Technology, Inc., Gwangju, Korea) equipped with a He–Ne laser (λ = 632.8 nm) and a gadolinium gallium garnet (GGG) prism (n = 1.965). Transmittance spectra of the obtained films were measured using a V-560 spectrophotometer (JASCO, Tokyo, Japan).

## 3. Results and Discussion

### 3.1. Characterization of Titanium Oxide Nanoparticles

The TiNPs (TiNP-1, 2, and 3) used in this study were characterized by DLS, XRD, and TEM analyses (Figure 1). TEM images of TiNPs indicated that all TiNPs were nano-sized particles, however, the morphologies were different. The average sizes of TiNP-1, TiNP-2, and TiNP-3, determined by DLS measurements, were 40 nm, 20 nm, and 70 nm, respectively. These sizes were different from the sizes observed in the TEM images (average size of TiNPs were 36 nm, 27 nm (40 nm 12 nm for major and minor axes respectively), and 9 nm respectively), indicating that the TiNPs had coagulated in the aqueous dispersions. XRD patterns of all TiNPs used in this study exhibited high intensity peaks at 25°, 47°, and 55°, indicating that the TiNPs were predominantly in the anatase phase. All peaks were in good agreement with the standard spectrum (JCPDS no.: 88-1175 and 84-1286 (anatase)). All peaks were broad as compared with those of the micro-sized anatase phase titanium oxide crystal, and some small, unidentifiable shoulders were observed. These results suggest that the TiNPs were composed of irregular polycrystalline and amorphous phases.

### 3.2. Preparation of Hydrophilic Polymer Composite Films Containing TiNP

Conventional hydrophilic polymers such as poly(acrylic acid) (*p*AA), poly(ethylene glycol) (*p*EG), poly(vinyl pyrrolidone) (*p*VP), and poly(vinyl alcohol) (*p*VA) were tested as matrix polymers for comparison with *p*HEAAm. Aqueous mixtures of TiNP-1 with the hydrophilic polymers were prepared at 70 wt% TiNP-1. As shown in Figure 2, the aqueous mixtures with *p*AA and *p*EG formed hydrogels at this mixing ratio, and the aqueous mixture with *p*VP formed a viscous solution. No gelation was observed in TiNP-1 with *p*VA and *p*HEAAm, indicating that the hydroxyl groups in the polymer side chains increased the compatibility of the polymer with TiNP-1. Furthermore, the transparency of the aqueous mixture with *p*HEAAm was higher than that of the other mixtures. These results suggest that *p*HEAAm is a good dispersant for TiNP-1 in water. As *p*HEAAm possesses a relatively high number of hydrogen bonding sites, such as hydroxyl and amide groups, it is reasonable to postulate that they contribute to the dispersion of TiNP-1 via hydrogen bonding with the surface of titanium oxide.

Aqueous mixtures of TiNP/polymer composites (300 µL) were dropped onto glass plates and dried at 10 °C to obtain thin films. It was difficult to obtain thin films of the aqueous mixtures with *p*AA, *p*EG and *p*VP, due to their high viscosity. Yellow, crack-free thin films were obtained from the aqueous mixtures with *p*VA and *p*HEAAm. The *p*HEAAm-based thin film prepared had higher transparency compared with *p*VA-based thin film. The TiNP-1/*p*HEAAm composite films were used for investigation of the refractive indices of the composite thin films.

### 3.3. Compatibility of TiNPs with the pHEAAm Polymer

The *p*HEAAm composite films containing different quantities of TiNP (50, 60, 70, 80, 90 and 95 wt%) were prepared by drop casting the aqueous mixtures onto a glass plate, followed by drying at ambient pressure and temperatures of 10, 25 or 50 °C. At the relatively low TiNP concentrations of 50 and 60 wt%, transparent yellow films were obtained from the aqueous mixtures of TiNP-1/*p*HEAAm and TiNP-2/*p*HEAAm, and transparent colorless films were obtained from mixtures of TiNP-3/*p*HEAAm. However, at higher TiNP concentrations, cracks and blurs were observed in the polymer composite films. TiNP-1 exhibited better compatibility with *p*HEAAm than TiNP-2 and TiNP-3. In the case of TiNP-1/*p*HEAAm, transparent and crack-free films were obtained at each temperature (10, 25 or 50 °C) for concentrations of up to 90 wt%, 80 wt% and 70 wt%, respectively. Figure 3 shows typical photos of TiNP-1/*p*HEAAm composite films with a variety of composition ratios, on glass plates, prepared at 10 or 50 °C. The lower drying temperature (10 °C) was suitable for the formation of TiNP/*p*HEAAm composite films with crack-free surfaces, on the glass plates. As a result, transparent and crack-free films were obtained with TiNP-1, TiNP-2, and TiNP-3 for concentrations of up to 90 wt%, 70 wt% and 60 wt%, respectively. At higher concentrations, cracks were observed on the surfaces of films prepared, even at a drying temperature of 10 °C (Figure 4).

### 3.4. Refractive Indices of the TiNP/pHEAAm Composite Films

The refractive indices (*n*) of the polymer composite films were measured using a prism coupling refractometer, and the results are summarized in Table 1. The refractive index of the TiNP/*p*HEAAm composite films increased as the proportion of TiNP increased. The highest refractive index observed in this study was 1.90, which was obtained for a mixture of *p*HEAAm with TiNP-1 at 90 wt%. The n value was significantly improved compared with the n value of the *p*HEAAm film (*n* = 1.53) without TiNP.

The observed n values were plotted against the concentration of TiNP (Figure 5) and compared with n values (dotted lines) calculated using Equation (1) and refractive indices of 2.50 for titania NPs and 1.53 for *p*HEAAm [30].
(1)$$\frac{n^{2}-1}{n^{2}+2}=(1-c)\frac{\rho }{\rho_{1}}\frac{n_{1}^{2}-1}{n_{1}^{2}+2}+c\frac{\rho }{\rho_{2}}\frac{n_{1}^{2}-1}{n_{1}^{2}+2}$$
where *n*, *n*_1_, and *n*_2_ are the refractive indices of the composite, polymer (*p*HEAAm), and TiNP (TiNP-1), respectively, and *ρ*, *ρ*_1_, and *ρ*_2_ are the densities of the composite, polymer (*p*HEAAm), and TiNP (TiNP-1), respectively, and *c* is the concentration (wt%) of TiNP (TiNP-1).

The refractive indices of the thin films on the glass plates were measured using a prism coupler equipped with a HeNe laser and a GGG prism (*n* = 1.965). As shown in Figure 5, the observed n values of the TiNP-1/*p*HEAAm composites prepared at 50 °C were in good agreement with the theoretical values for TiNP-1, calculated with 0% impurity (Cal-A (1.52 (*n*_1_), 4.29 (*n*_2_), 1.11 g cm^−3^ (*ρ*_1_), and 4.23 g cm^−3^ (*ρ*_2_) were used for calculation), green line in Figure 5a), in the TiNP-1 composition range from 50 to 70 wt% TiNP-1, but lower at a TiNP-1 composition of 80 wt%. In contrast, the observed n values of the TiNP-1/*p*HEAAm composites prepared at 10 °C were lower than Cal-A in the TiNP-1 composition range from 50 to 90 wt%. As shown in Figure 5b, the thermogravimetric analysis indicated that the solid components in the aqueous TiNP-1 contained 9.6 wt% of evaporative and/or flammable impurities. The lower values for the experimentally obtained refractive indices compared with Cal-A is probably due to contamination by organic impurities. The Cal-B (1.52 (*n*_1_), 4.29 (*n*_2_), 1.47 (RI of unknown), 1.11 g cm^−3^ (*ρ*_1_), 4.23 g cm^−3^ (*ρ*_2_), 1.0 g cm^−3^ (density of unknown), and 9.6 wt% (composition of unknown) were used for calculation, blue broken line in Figure 5a) indicates the calculated n values with 9.6 wt% of impurities (for calculation, 1.0 g cm^−1^ and 1.47 were used for the density and the refractive index of impurity) added, and is in good agreement with the experimental n values of the composite films prepared at 10 °C. The experimental n value of the composite film at the higher TiNP-1 composition of 95 wt% was also significantly lower than the calculated value. As described in Section 3.2, cracks were observed on the surface of the composite films with higher TiNP-1 concentrations, whereas no cracks formed on films with lower TiNP-1 concentrations. The decreased n value at the higher concentration of TiNP-1 was probably due to the formation of cracks on the surface of the composite film.

The transmittance UV-Vis spectra of composite films containing TiNP-1 at different concentrations were measured (Figure 6). Although the transmittance of the composite film gradually decreased with increasing TiNP-1 concentration, the composite film containing 90 wt% of TiNP-1 exhibited more than 80% transmittance in the wavelength range from 480 to 900 nm. Owing to absorption of the original TiNP-1 at wavelengths below 480 nm, the transmittances of the composite films decreased below 480 nm. It should be noted that the composite films were yellow but transparent. The yellow color was caused by the original TiNP-1 dispersion, which was probably due to the impurity; therefore, a colorless film may be obtained by removing the impurities from the TiNP dispersion.

## 4. Conclusions

In this study, various hydrophilic polymers with hydrogen bonding sites such as *p*HEAAm, poly(vinyl alcohol), poly(ethylene glycol), and poly(acrylic acid) were examined for the preparation of the polymer/titanium oxide nanoparticle composite films with high refractive indices. Among the hydrophilic polymers, *p*HEAAm exhibited the best compatibility with the titanium oxide nanoparticles. The compatibility was also affected by the nature of the titanium oxide. Further investigations into the hybridization behavior of *p*HEAAm with TiNP are needed, but the comparably higher compatibility of *p*HEAAm with titanium oxide nanoparticles is probably due to the formation of hydrogen bonding between the amide and hydroxide groups of *p*HEAAm, and the surface of titanium oxide nanoparticles. In the combination of *p*HEAAm and TiNP-1 (40 nm), the transparent composite thin films were obtained by casting the aqueous mixture of them at 10 °C even at the high concentration of 90 wt%. The experimental refractive indices were in good agreement with the calculated values, indicating that the TiNP-1 was homogeneously dispersed in the *p*HEAAm polymer matrix. The refractive index of the thin films was dropped at 95 wt% of TiNP-1. The composite thin films exhibited excellent refractive indices, with the highest refractive index reaching a value of 1.90. Laser microscopic observations indicated that the surface of the thin films was smooth and crackless at 90 wt% of TiNP-1 or less, but the cracks were observed on the surface of the thin films composed of TiNP-1 of more than 90 wt%. Thin films with high refractive indices are useful in a wide range of applications such as photo-devices, coatings, lighting equipment, and lenses.

## Figures and Tables

**Figure 1 nanomaterials-09-00514-f001:**
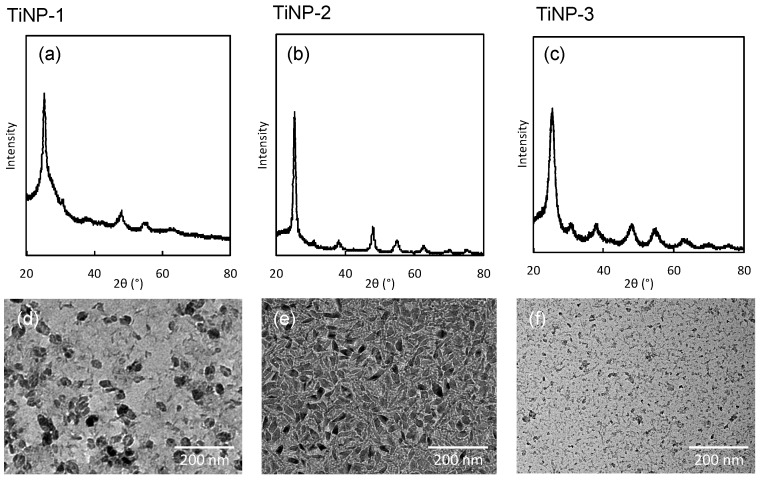
XRD patterns (**a**–**c**) and TEM images (**d**–**f**) of titanium oxide nanoparticles. TiNP-1 (**a**,**d**), TiNP-2 (**b**,**e**), and TiNP-3 (**c**,**f**).

**Figure 2 nanomaterials-09-00514-f002:**
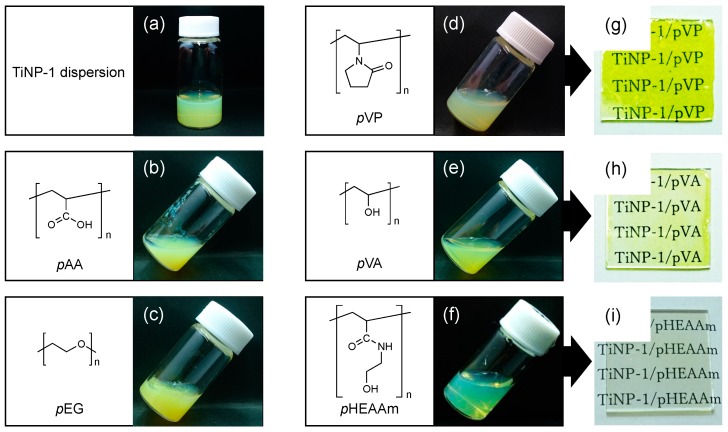
Photos of the aqueous dispersion of TiNP-1 (**a**), aqueous mixtures of TiNP-1 and the water-soluble polymers ((**b**): *p*AA, (**c**): *p*EG, (**d**): *p*VP, (**e**): *p*VA, (**f**): *p*HEAAm), and the composite films ((**g**): *p*VP, (**h**): *p*VA, (**i**): *p*HEAAm) prepared by casting of the aqueous mixture on a glass plate at 10 °C.

**Figure 3 nanomaterials-09-00514-f003:**
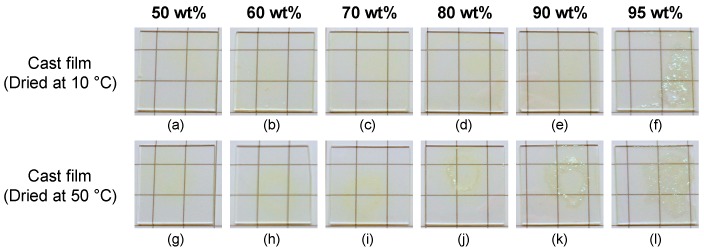
Photos of poly(*N*-hydroxyethyl acrylamide) (*p*HEAAm)-based thin films with different compositions of TiNP-1 on glass plates ((**a**,**g**): 50 wt%, (**b**,**h**): 60 wt%, (**c**,**i**): 70 wt%, (**d**,**j**): 80 wt%, (**e**,**k**): 90 wt%, (**f**,**l**): 95 wt%). The thin films were prepared by drop casting of the mixtures onto the glass plates and dried at 10 °C (**a**–**f**) or 50 °C (**g**–**l**).

**Figure 4 nanomaterials-09-00514-f004:**
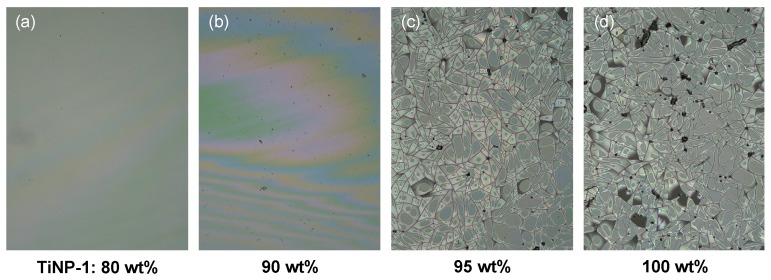
Laser microscopic images (120×) of the surfaces of TiNP-1/*p*HEAAm composite films with different compositions. (**a**) TiNP-1: 80 wt%, (**b**) TiNP-1: 90 wt%, (**c**) TiNP-1: 95 wt%, (**d**) TiNP-1: 100 wt%. The composite films were prepared by casting at 10 °C.

**Figure 5 nanomaterials-09-00514-f005:**
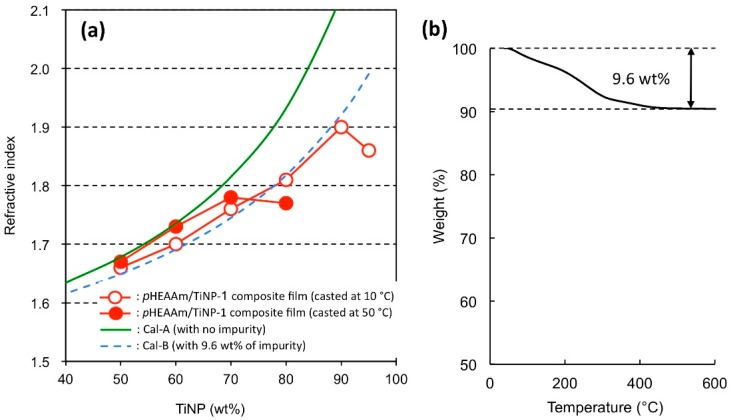
(**a**) Refractive indices of TiNP-1/*p*HEAAm composite films under a variety of preparation conditions and (**b**) thermogravimetric (TG) analysis of dried powder prepared from TiNP-1.

**Figure 6 nanomaterials-09-00514-f006:**
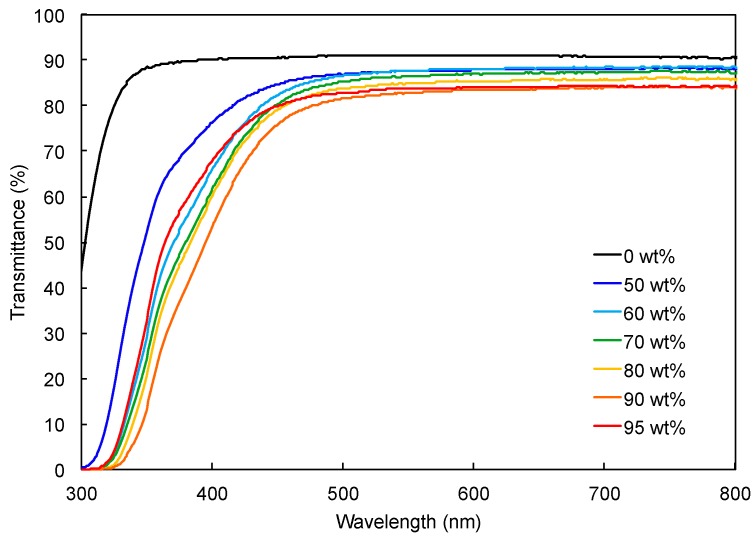
UV-Vis spectra of TiNP-1/*p*HEAAm composite films with different concentrations of TiNP-1. The composite films were prepared by drop casting the aqueous mixture of TiNP-1 and *p*HEAAm on a glass plate and drying at 10 °C.

**Table 1 nanomaterials-09-00514-t001:** Refractive indices of TiNP/*p*HEAAm composite films under a variety of preparation conditions.

Composition TiNP:*p*HEAAm		TiNP-1		TiNP-2		TiNP-3	
	10 °C ^a^	25 °C ^a^	50 °C ^a^	10 °C ^a^	50 °C ^a^	10 °C ^a^	50 °C ^a^
95:5	1.86	-	-	-	-	-	-
90:10	1.90	1.79	-	-	-	-	-
80:20	1.81	1.78	1.77	-	-	-	-
70:30	1.76	1.75	1.78	1.74	1.70	-	-
60:40	1.70	1.72	1.73	1.71	1.64	1.78	-
50:50	1.66	1.67	1.67	1.67	1.61	1.72	1.72
0:100	1.53	-	-	1.53	-	1.53	-

^a^ Drying temperature of TiNP/*p*HEAAm composite film.

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
