# Peer review of "Preparation of High Refractive Index Composite Films Based on Titanium Oxide Nanoparticles Hybridized Hydrophilic Polymers"

_nanomaterials, 2019, doi:10.3390/nano9040514_

Round 1
Reviewer 1 Report
In this manuscript, the authors fabricated thin film with high refractive index by mixing TiO2 NPs and hydrophilic polymers. The effect of the type of polymers, and the weight ratio between NPs and polymers on refractive index was systematically studied. Overall, the manuscript is well organized thus I can recommend its publication in the journal after considering the following minor comment.
1. Were all TiNPs prepared by using same ligands? I am wondering whether such different surfactants used during the synthesis of NPs impact the optical properties of resulting composites film or the compatibility of NPs with polymers?
2. In line 100, please specify the value of TiNP size observed by TEM. Even though authors claim that TiNP-3 is the largest one, its TEM image in Figure1 looks different. Please check it.
3. In line 135, Why is the selected drying temperature is either 10 or 50?
4. Typos: In line 181-182, please mark subscript on “n,n1, and n2” and “p,p1, and p2”.
Reviewer 2 Report
This work is worth of publication in Nanomaterials without further changes.
Reviewer 3 Report
Comments:
The current manuscript deals with the effect of changes in refractive index hybrid thin films based on titanium oxide nanoparticles in various hydrophilic polymers. It is interesting, however, the manuscript cannot accept without changes. Below the authors will find some comments:
The title did not reflect the whole work of the study. The authors investigated various hydrophilic polymers, however in the title only one polymer is mentioned.
The abstract did not reflect suitably the study and has to be reworked. There are only results (more or less).
The introduction should be also reworked. It is mainly a listing of previous study and investigated materials however, not a comprehensive presentation about the previous results from other scientific studies.
There was only briefly and too generally description of the research question.
Figure 4, Please mention also the drying temperature
Page 5 line 170: Please change the term “To the best of our knowledge”
Page 6: please change the Figure 5 and the description in the text “green line” and “blue line”, in the case that the publication will be printed in black and white by the reader.
The TG analysis (Figure 5b) was not described in the materials and method section.
The conclusion did not reflect the whole study and is more or less a summary. Therefore, this section has to reworked. Please do not use the term “…. will be reported elsewhere.”
Round 2
Reviewer 3 Report
The authors reworked the manuscript according to the recommendations of the reviewers. The paper is now fine, however, the title should still be revised to reflect the whole work in this study, e.g. High refractive index hybrid thin films based on titanium oxide nanoparticles dispersed in various hydrophilic polymer matrixes
